# Estimation of the fraction of COVID-19 infected people in U.S. states and countries worldwide

**Jungsik Noh**[ORCID]*, **Gaudenz Danuser**

Lyda Hill Department of Bioinformatics, University of Texas Southwestern Medical Center, Dallas, Texas, United States of America

* jungsik.noh@utsouthwestern.edu

## Abstract

Since the beginning of the coronavirus disease 2019 (COVID-19) pandemic, daily counts of confirmed cases and deaths have been publicly reported in real-time to control the virus spread. However, substantial undocumented infections have obscured the true size of the currently infected population, which is arguably the most critical number for public health policy decisions. We developed a machine learning framework to estimate time courses of actual new COVID-19 cases and current infections in all 50 U.S. states and the 50 most infected countries from reported test results and deaths. Using published epidemiological parameters, our algorithm optimized slowly varying daily ascertainment rates and a time course of currently infected cases each day. Severe under-ascertainment of COVID-19 cases was found to be universal across U.S. states and countries worldwide. In 25 out of the 50 countries, actual cumulative cases were estimated to be 5–20 times greater than the confirmed cases. Our estimates of cumulative incidence were in line with the existing seroprevalence rates in 46 U.S. states. Our framework projected for countries like Belgium, Brazil, and the U.S. that ~10% of the population has been infected once. In the U.S. states like Louisiana, Georgia, and Florida, more than 4% of the population was estimated to be currently infected, as of September 3, 2020, while in New York this fraction is 0.12%. The estimation of the actual fraction of currently infected people is crucial for any definition of public health policies, which up to this point may have been misguided by the reliance on confirmed cases.

## Introduction

Since its initial spread in China in December 2019, the coronavirus disease 2019 (COVID-19) has caused more than 860,000 confirmed deaths all over the world as of September 3, 2020 [1], and it continues to threaten the whole population most of which remain susceptible to infection by the severe acute respiratory syndrome coronavirus-2 (SARS-CoV-2). As an effort to contain the virus, the daily counts of laboratory-confirmed cases and deaths have been publicly reported in real-time [2]. However, substantial undocumented infections have obscured the

**Data Availability Statement:** All code, daily updated estimates, and their visualizations are freely available at a GitHub repository (https://github.com/JungsikNoh/COVID19_Estimated-Size-of-Infectious-Population).

**Funding:** This work was supported by Lyda Hill Philanthropies. The funders had no role in study design, data collection and analysis, decision to publish, or preparation of the manuscript.

**Competing interests:** The authors have declared that no competing interests exist.

actual fraction of at least once infected people. A computational study estimated the ratio of confirmed cases to actual cases, i.e., an ascertainment rate, to be only 14% during the early outbreak in China [3]. Large-scale seroprevalence studies aimed to estimate the actual number of infections and found severe under-ascertainment in several U.S. states, where the ascertainment rates varied from 4.2% in Missouri to 8.9% in New York and 16.7% in Connecticut until March or April 2020 [4, 5]. A recent nationwide study estimated that only 9.2% of actual infections were laboratory-confirmed in the U.S. until July 2020 [6]. More importantly, we still do not know how many individuals are currently infected in many countries and regions. The currently infected population is the cause of future infections and deaths. Its actual size in a region is a crucial variable required when determining the severity of COVID-19 and building strategies against regional outbreaks.

The daily counts of confirmed COVID-19 cases and deaths alone possess incomplete information on the relative abundance of epidemiological compartments of a population that is susceptible, infected, recovered, or deceased. Whether to be confirmed or not adds another layer of complexity to the categories of infected, recovered, or deceased compartments (Fig 1). In addition to the under-ascertainment, several limitations in the reported data make it challenging to estimate the number of currently infected cases: recovery events are not tracked in most countries; there are time delays between infection onset and laboratory-confirmation [3]; and even the death tolls are suggested to be under-reported in some regions [7, 8].

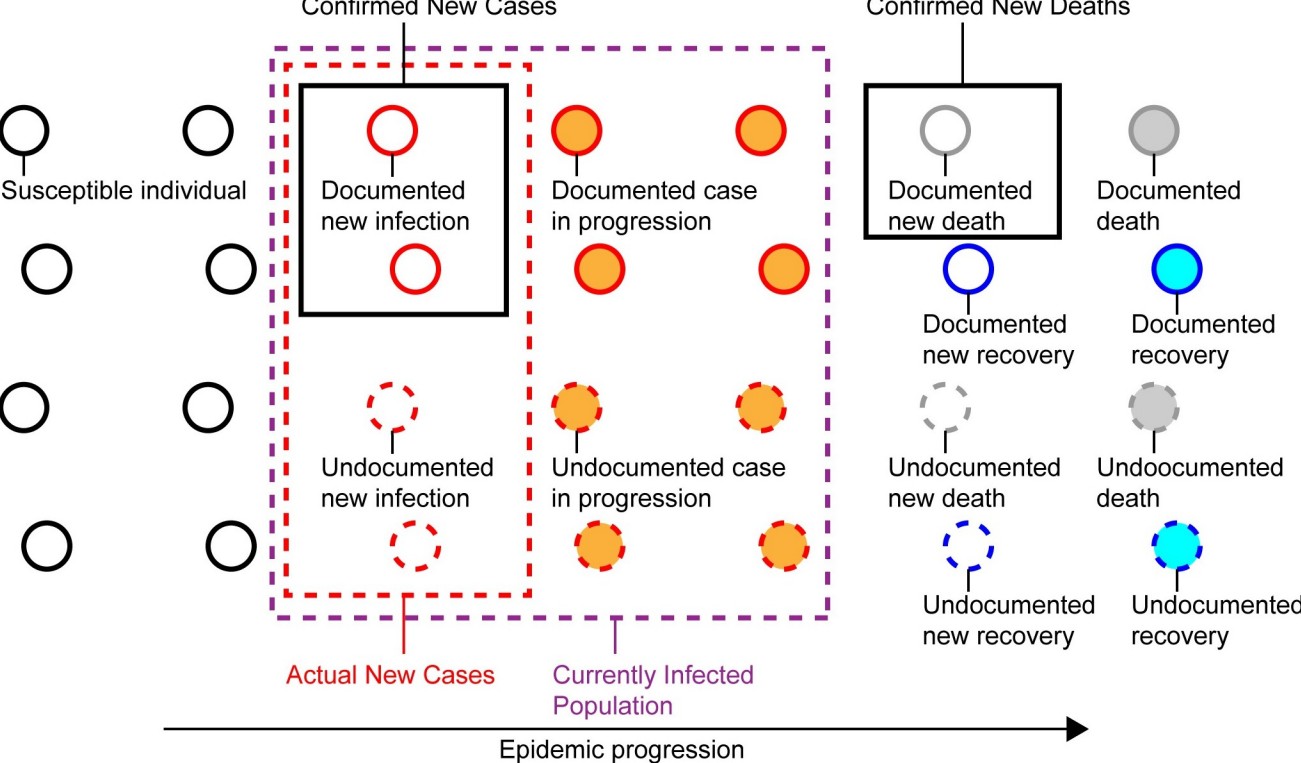

**Fig 1. Undocumented COVID-19 cases.** In an epidemic process, a population is categorized into susceptible, infected, deceased, or recovered individuals. Counts of confirmed COVID-19 cases, deaths, and recoveries are insufficient to calculate the number of currently infected individuals (purple dotted box) because of substantial undocumented infections not captured by diagnostic tests. The input to the proposed framework is the daily counts of confirmed new cases and deaths (black boxes). Using pandemic parameters such as the Infection-Fatality-Rate and the mean duration periods from infection to death and recovery, the framework estimates the counts of actual new cases (red dotted box) and currently infected individuals.

Key epidemiological parameters such as the Infection-Fatality-Rate (IFR) give us a clue to fill the gap between confirmed and actual infections, under the assumption that the number of undocumented deaths is negligible (Fig 1). The IFR of COVID-19 has been a focus of intensive research, yet studies from different locations and times have not reached a consensus estimate [9]. A recent large seroprevalence study in 133 cities of Brazil presented an IFR estimate of 1.0% [10]. In a different approach, a study analyzed early pandemic data in China combined with the prevalence obtained by PCR-testing of the entire international resident population repatriated from China. The authors' estimate of the IFR was 0.66% with a wide band of uncertainty (0.39%–1.33%, 95%-confidence interval) [11]. The same study also reported the mean duration from onset of symptoms to death or recovery (17.8 and 24.7 days, respectively) based on individual-level data.

This study presents machine learning-based estimates of actual sizes of currently infected populations in select countries and all 50 U.S. states. These fractions of infected people are derived by estimating daily ascertainment rates and subsequently adjusting the under-reported COVID-19 cases. The estimates are based on publicly available datasets of daily confirmed cases and deaths, and published estimates of key pandemic parameters. Using the proposed pipeline, an online repository presents visualizations of daily updates on the estimated actual fraction of infected people for the 50 countries with the most confirmed cases and for all 50 U.S. states [12].

## Methods

For this computational study, we used the dataset of confirmed cases and deaths for countries taken from the repository by the Center for Systems Science and Engineering (CSSE) at Johns Hopkins University [2], and the dataset for U.S. states taken from the COVID Tracking Project [13].

To infer the actual number of infections across countries and regions, we utilized the epidemiological estimates of the IFR and the mean duration from the symptom onset to death or recovery presented by [11]. The IFR is known to heavily depend on age groups [11] and would vary across countries with different age distributions. Therefore, applying the above IFR estimate to a region with an extremely young or old population will be inappropriate. But, considering the estimate's large estimation uncertainty as shown in the above, the confidence interval is expected to cover the true IFRs of most countries and U.S. states.

Our computational pipeline started with initial estimates of time courses of actual new infections and new recoveries, derived from the daily confirmed deaths, the IFR estimate, and the mean duration from infection to death and recovery (S1 Fig). The estimated new infections led to two other initial estimates: a daily ascertainment rate that is the ratio of confirmed new infections to the estimated new infections, and the number of currently infected cases each day. Then a regression model was applied to find a functional relation of the daily infected cases to the daily ascertainment rates, accounting for a common temporal trend of under-reporting shown in both of the daily ascertainment rate and the ratio of confirmed cases to infected cases. Employing the expectation-maximization (EM) algorithm, the pipeline iteratively updated the time courses of ascertainment rates, new infections, and currently infected cases based on each other until convergence to obtain final estimates. The same EM iterations were applied with the lower/upper limits of the IFR estimate to obtain upper/lower 95%-confidence limits of the estimated number of infections, respectively (S1 Fig, See S1 Appendix for method details).

The estimates of the actual number of infections were validated using seroprevalence data from the large-scale surveys conducted by the Centers for Disease Control and Prevention

(CDC), New York state, and a recent nationwide study [4–6, 14]. The surveys collected blood samples in multiple U.S. states and tested for antibodies to SARS-CoV-2 to estimate the proportion of people who were previously infected. The CDC, New York state and the recent study presented statewide seroprevalence estimates for five, one, and 46 U.S. states, respectively. The seroprevalence rates at different times were compared with the computationally estimated cumulative incidence rates on the date one-week prior to the mid-points of the blood collection periods, accounting for time delays from infection to antibody detection.

## Results

In comparison with the seroprevalence in six U.S. states presented by the CDC and New York state, overall the described pipeline yielded accurate estimates of cumulative incidence, with the exception of Utah (Fig 2A). The estimated cumulative incidence in New York by April 17, was 9.6% (5.2%–15.7%), which was in line with a seroprevalence of 14.0% (NYS), although the estimate had a large uncertainty originating in the wide confidence interval of the IFR estimate. The cumulative incidence rates in Washington state were 0.6% (0.5%–0.7%) and 1.9% (1.2%–2.9%) by March 21 and April 27, respectively, which were close to the seroprevalence of 1.1% and 2.1% surveyed in western Washington state measured one week later. The estimated incidence rate of 3.7% (1.9%–6.3%) in Connecticut by April 23, was in line with the seroprevalence of 4.9% in the first round, while the incidence estimate of 12.3% (6.1%–20.8%) by May 17, differed from a seroprevalence of 5.2% in the second round. As some studies reported that the antibodies decreased over time in some patients [15, 16], the second round seroprevalence rates in the four states seemed to be unstable. Indeed, the rates became even smaller than the first round in Utah and remained almost the same in Connecticut and Missouri. In comparison to the seroprevalence rates in the first round in Louisiana and Missouri, the cumulative incidence rates seemed to be slightly underestimated. In Utah, the estimates and seroprevalence rates showed prominent discrepancy, where the significantly low incidence estimate by April 20 suggested a possibility of under-reported death tolls.

The recent nationwide serologic survey tested for SARS-CoV-2 antibodies in randomly sampled patients receiving dialysis during July 2020, using a more accurate antibody test [6]. Given that both of seroprevalence estimates and our cumulative incidence estimates showed wide 95%-confidence intervals ranging ~10%, the estimated cumulative incidence rates were in line with the seroprevalence rates in 45 states and Washington D.C. with few exceptions such as New York and New Jersey (Fig 2B). The two measures of actual infection showed a Pearson correlation of 73% (P-value < 0.0001). Since our estimates are based on the assumption that the number of deaths is accurate, the under-estimated incidence rate for New York by July 8, 2020 (19.4%) compared to the seroprevalence (33.6%) supported a previous report of considerably under-reported COVID-19 deaths in New York [7].

Applied across countries and U.S. states, the proposed framework estimated actual time courses of new infections and currently infected cases. In early April, the U.S. reported ~30,000 daily confirmed cases. In striking contrast, the proposed estimation suggested a number of actual daily cases of more than 400,000, showing that the daily ascertainment at that time was less than 10% (Fig 3A). As of September 3, 2020, 0.9% (0.5%–1.6%) of the U.S. population was estimated to be currently infected. In Brazil, the under-reporting was also severe early in the pandemic, and only gradually improved over time, as in the U.S. As a result, the peak in actual daily cases seemed to have occurred between June 1 and June 8, 2020, reaching nearly 250,000 cases in contrast to ~25,000 confirmed daily cases (Fig 3B). This time of the peak in new infections was earlier than the peak in confirmed cases, which fell between July 27 to August 3. The currently infected cases in Brazil were estimated to be 2.3% (1.1%–3.9%) of

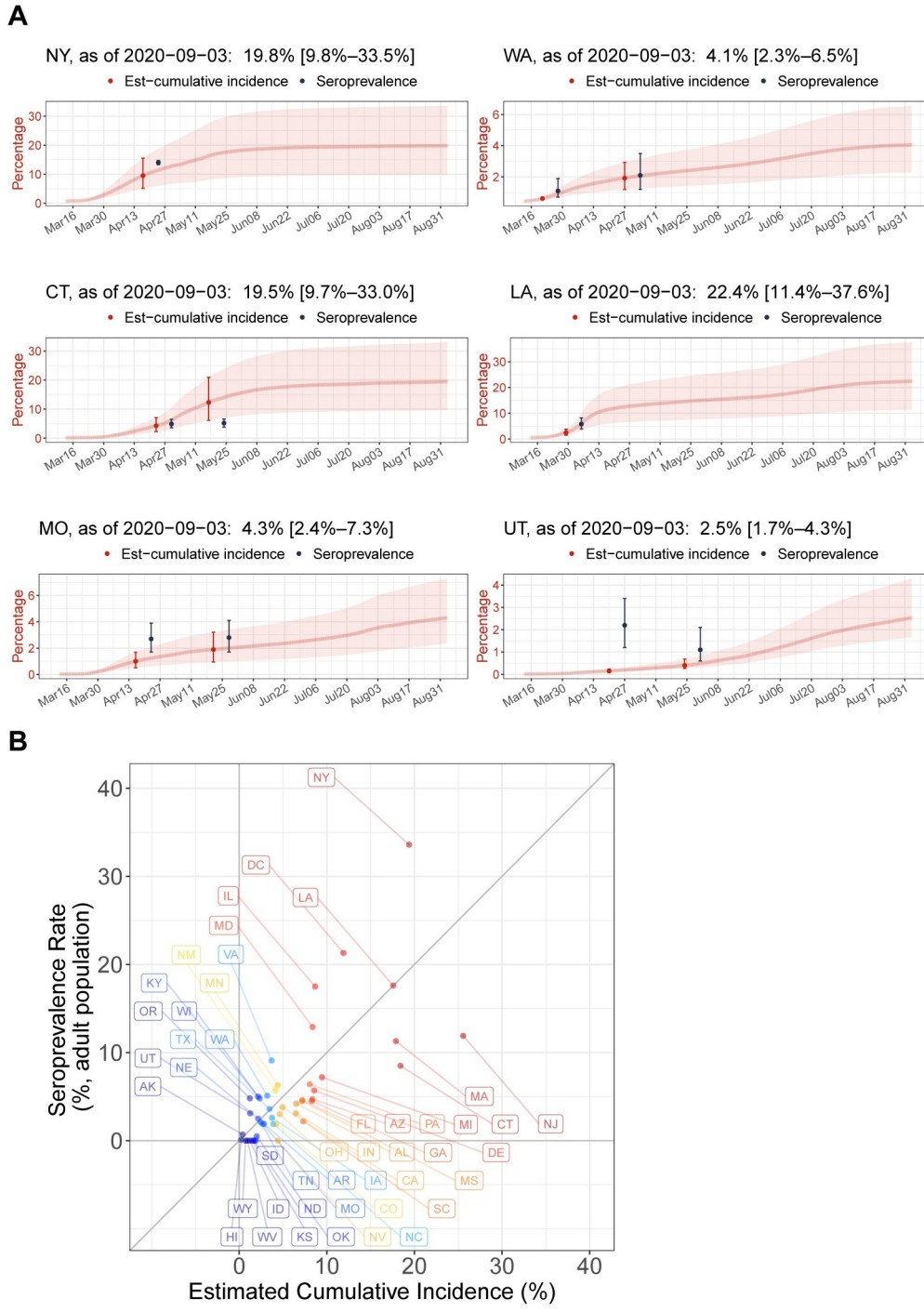

**Fig 2. Validation of prediction framework using seroprevalence rates in U.S. states. (A)** Seroprevalence rates in six U.S. states (black) surveyed until May 2020, are overlaid on computationally estimated time courses of cumulative incidence rates (red) from March 13 to September 3, 2020, for New York, Washington state, Connecticut, Louisiana, Missouri, and Utah from upper-left to lower-right. The indicated date of the seroprevalence rate is the mid-point of the serum collection period. The corresponding cumulative incidence estimate is on the date one-week prior to the date of the seroprevalence rate to account for time delays from infection to antibody detection. Error bars and shaded bands indicate 95% confidence intervals. (**B**) The Y-axis shows the seroprevalence rates in adult (≥18 years) populations of 45 U.S. states and Washington D.C. estimated from a nationwide plasma sample (n = 28,503) of patients on dialysis during July 2020. The X-axis shows the computationally estimated cumulative incidence rates for the states on July 8, 2020, that is one week prior to the mid-point of the plasma sample collection period, July 2020.

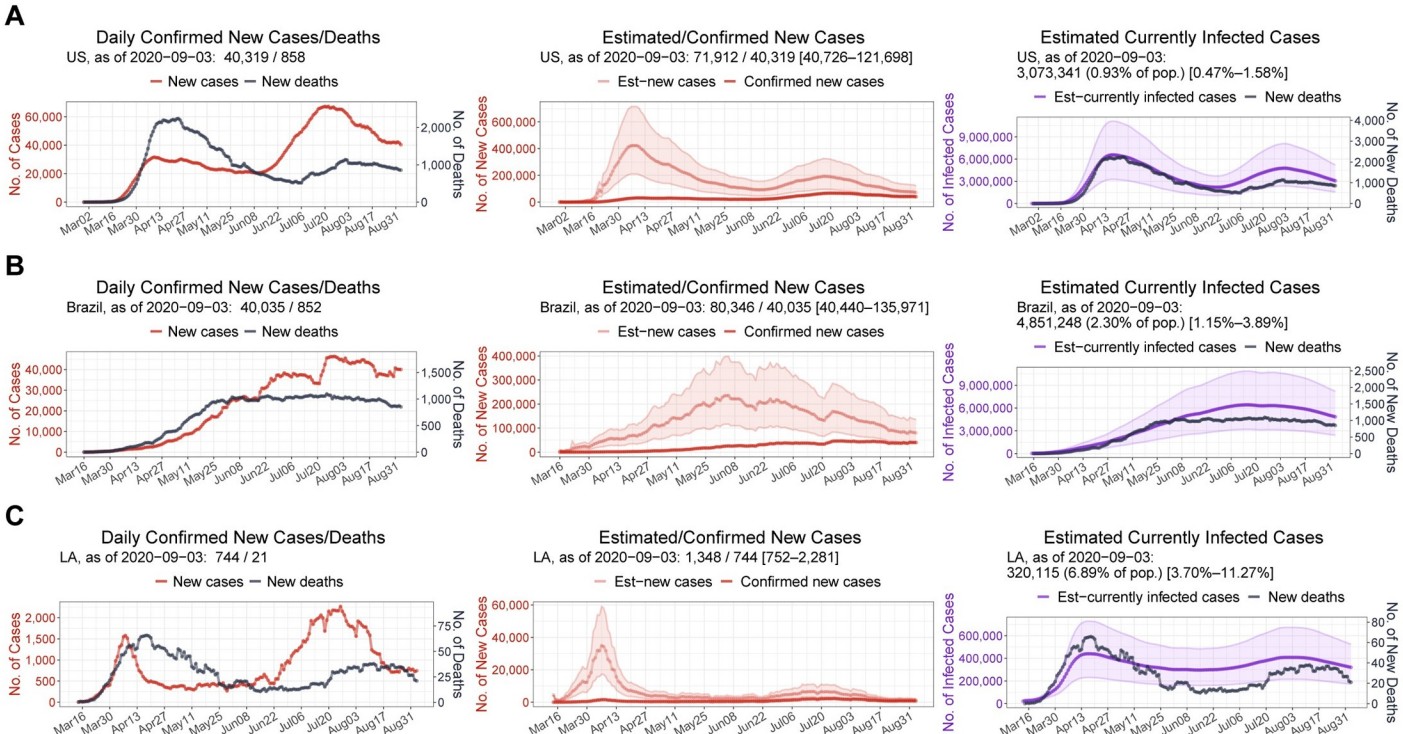

**Fig 3. Estimated time courses of actual new cases and current infections.** 7-day rolling-averaged counts of daily confirmed new cases and deaths (left) until September 3, 2020, for the U.S. (**A**), Brazil (**B**), and Louisiana (**C**). An estimate of new cases (middle) is the under-reporting-adjusted number of newly infected individuals each day. An estimate of current infections (right) is the under-reporting-adjusted number of infected individuals who have not yet been recovered or deceased. Shaded bands indicate 95%-confidence intervals.

the total population as of September 3, 2020. Among U.S. states, Louisiana showed the highest estimated fraction of currently infected people, 6.9% (3.7%–11.3%) as of September 3, 2020 (Fig 3C). The first peak in the daily new cases in Louisiana was ~1,500 around April 6, but the actual new cases at that time were estimated to be already more than 30,000, indicating the severity of under-reporting in Louisiana during the month of April.

The severe under-ascertainment was universal across the 50 countries with the most confirmed cases and 50 U.S. states. The ascertainment rates for the whole period until September 3, 2020, widely varied from 5% in Italy to 99% in Qatar, and from 8% in Connecticut to 71% in Alaska (Fig 4A). Among them, 25 countries, 19 U.S. states, and Washington D.C. showed an ascertainment rate less than 20% for the entire time of the pandemic. Focusing on only the past two weeks the ascertainment rates unfortunately have not improved much in these countries, while the recent rates of U.S. states increased overall as of September 3, 2020. Interestingly, many of the countries with high ascertainment rates from the beginning of the outbreak were the ones that previously experienced Middle East respiratory syndrome coronavirus.

The under-reporting adjustment allowed us to monitor the actual severity of the virus spread across countries and U.S. states, and especially the estimated sizes of currently infected populations helped to identify fast-changing COVID-19 hotspots. In Peru, Belgium, and Brazil, more than 10% of the population were estimated to be once infected as of September 3, 2020 (Fig 4B). Across U.S. states, the cumulative incidence rates ranged from 28.2% (14.0%–47.7%) in New Jersey to 0.9% (0.6%–1.5%) in Hawaii (Fig 4B). As of September 3, 2020,

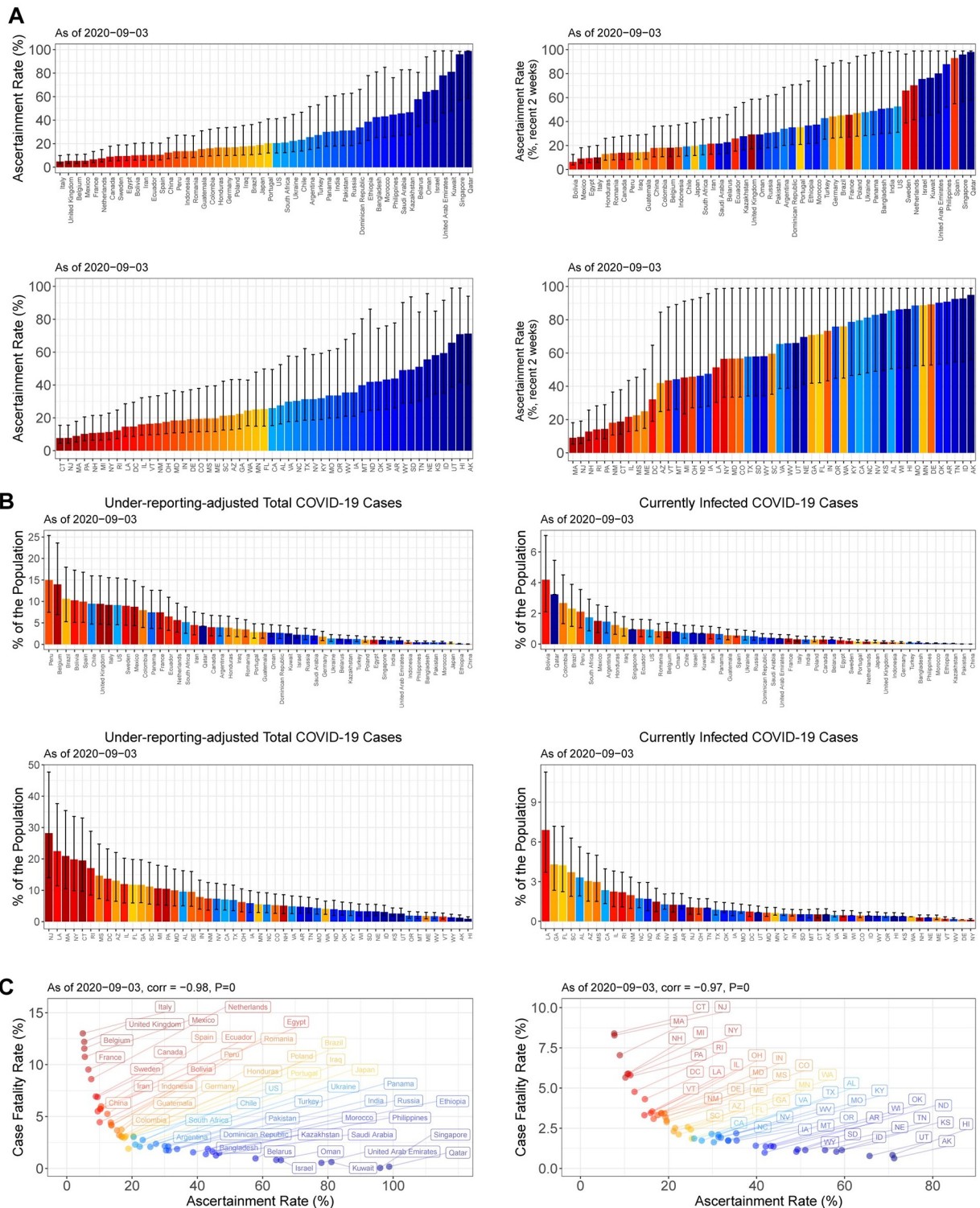

**Fig 4. Estimates of ascertainment rates, cumulative incidence rates, and actual fractions of current infections in 50 countries and 50 U.S. states.** (**A**) Estimates of ascertainment rates for the whole period until September 3, 2020 (left), and recent ascertainment rates (August 21–September 3, 2020) (right), in 50 countries with the most confirmed cases (upper) and 50 U.S. states (lower). (**B**) Cumulative incidence rates (left), and percentages of currently infected individuals in each population (right) in the 50 countries (upper) and 50 U.S. states (lower). Error bars indicate 95%-confidence intervals. (**C**) Scatter plots between the crude case-fatality-rates and the ascertainment rates for the 50 countries (left) and 50 U.S. states (right). Spearman rank correlations and their P-values are shown.

COVID-19 hotspots among U.S. states were estimated to be Louisiana, Georgia, and Florida, where currently infected cases were estimated to be more than 4%.

The estimated fractions of current infections differentiated New Jersey from New York, both of which experienced severe early outbreaks (Fig 4B). The confirmed new cases per 100,000 population were 3.8 in New Jersey and 3.7 in New York as of September 3, 2020, suggesting that the virus spread was under control in both states. However, because of the differences in recent ascertainment rates between the two states (Fig 4A) the fractions of currently infected people were 1.05% (0.52%–1.78%) in New Jersey and 0.12% (0.06%–0.20%) in New York, as of September 3, 2020. This reveals New Jersey as still with a considerable infected population whereas New York has become one of the safest states.

Since the beginning of the COVID-19 pandemic, the Case-Fatality-Rates (CFRs) have displayed huge differences between countries, adding confusion to how deadly SARS-CoV-2 is. The crude CFRs, which are the ratios of total confirmed deaths to total confirmed cases, ranged from 13.0% in Italy to 0.05% in Singapore as of September 3, 2020. Our analysis now reveals that the variation on CFR reports across the countries and U.S. states is primarily associated with the massive differences in ascertainment rates between the locations (Fig 4C). The Spearman rank correlations between CFR and ascertainment rate were -98% and -97% (P-values $< 0.0001$) for the analyzed countries and the U.S. states, respectively. After adjustment for the under-reporting, the inferred IFRs, which were based on the assumed IFR 0.66%, did not correlate with the ascertainment rates (S2 Fig). Thus, a high CFR in a region is shown to be a result of severe under-reporting of the cases.

## Discussion

We presented machine learning-based estimates of daily counts of actual COVID-19 infections and currently infected cases across U.S. states and countries. Our cumulative incidence estimates were close to existing seroprevalence estimates for U.S. states with a few exceptions. In comparison with recently published seroprevalence rates for 46 U.S. states, our cumulative incidence estimates showed no systematic deviation from the seroprevalence, which indicated that the employed IFR estimate showed unbiased performance. Our analyses strongly supported the conclusion from the seroprevalence surveys, demonstrating that the severe under-ascertainment was universal across U.S. states and countries. In many regions, recent ascertainment rates were still low and our report showed how many infections should have been identified.

Unlike seroprevalence surveys, our computational approach provides daily updated estimates across U.S. states and countries worldwide. More importantly, our framework estimates the actual fraction of currently infected people in each region. To our knowledge this is the first model to provide this prediction. The estimated number of current infections can serve as an initial target in planning effective contact tracing. Since the developed pipeline requires simple input, it is widely applicable to more granular analyses of specific regions or communities, for which the number of confirmed cases and deaths are being tracked.

The proposed estimation heavily relies on the published estimate of the IFR, which is known to have a large uncertainty. Our estimates of actual cases would become more accurate if the IFR estimate were optimized to a specific region and its uncertainty could be reduced. Depending on available datasets in each region, the estimation of actual cases can be improved by augmenting more information such as daily positivity rates of diagnostic testing or daily hospitalized cases.

Estimating actual numbers of COVID-19 infections based on under-reported limited data has been a challenging task, especially since some regions display diverse dynamic patterns in

the infections and ascertainment rates. Therefore, the quality of the presented estimates may be poor for some U.S. states or countries. The plausibility of estimated time courses of currently infected cases can be assessed by daily rates of deaths among the infected people. A large variation in the daily death rates may indicate inaccuracy in the estimated time course. In the online repository since September 22, 2020, the estimation quality based on the daily death rates were annotated to indicate a few poor estimates among all the regions [12]. As the pandemic progresses, the pipeline would need to be adapted to the increasing complexity of the infection data.

In conclusion, this study demonstrates that severe under-ascertainment has obscured the true severity of widespread COVID-19 all over the world. In the majority of the 50 countries, actual cumulative cases were estimated to be 5–20 times greater than the confirmed cases. Given that the confirmed cases only capture the tip of the iceberg in the middle of the pandemic, the estimated sizes of current infections in this study provide crucial information to determine the regional severity of COVID-19 that can be misguided by the confirmed cases.

## Supporting information

**S1 Fig. Workflow to estimate time courses of actual infections.** (A) Expectation-maximization (EM) iteration to update latent time courses involved in actual infections. (B) Workflow of initialization, EM iterations, and calculation of confidence intervals.
(PDF)

**S2 Fig. Inferred infection-fatality rates and ascertainment rates.** Scatter plots between the inferred infection-fatality-rates (IFR) and the whole period ascertainment rates for the 50 countries (left) and 50 U.S. states (right). The inferred IFR is the ratio of total confirmed deaths to the under-reporting-adjusted total number of cases on a date 18-day before, accounting for the mean duration from infection to death. Spearman rank correlations and their P-values are shown.
(PDF)

**S1 Appendix. Supplementary methods.**
(PDF)

## Author Contributions

**Conceptualization:** Jungsik Noh.

**Formal analysis:** Jungsik Noh.

**Investigation:** Jungsik Noh.

**Methodology:** Jungsik Noh.

**Visualization:** Jungsik Noh.

**Writing – original draft:** Jungsik Noh, Gaudenz Danuser.

**Writing – review & editing:** Jungsik Noh, Gaudenz Danuser.

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
