## [Decision Letter · Decision Letter 0]

10 Dec 2020

PONE-D-20-32239

Estimation of the fraction of COVID-19 infected people in U.S. states and countries worldwide

PLOS ONE

Dear Dr. Noh,

Thank you for submitting your manuscript to PLOS ONE. After careful consideration, we feel that it has merit but does not fully meet PLOS ONE’s publication criteria as it currently stands. Therefore, we invite you to submit a revised version of the manuscript that addresses the points raised during the review process.

Your manuscript was reviewed by 2 experts in the field. The reviewers identified several important problems in your submission. Please consider the attached comments and provide point-by-point responses

We look forward to receiving your revised manuscript.

Kind regards,

Yury E Khudyakov, PhD

Academic Editor

PLOS ONE

Journal Requirements:

Reviewers' comments:

Reviewer's Responses to Questions

**Comments to the Author**

1. Is the manuscript technically sound, and do the data support the conclusions?

Reviewer #1: Yes

Reviewer #2: No

2. Has the statistical analysis been performed appropriately and rigorously? 

Reviewer #1: Yes

Reviewer #2: No

3. Have the authors made all data underlying the findings in their manuscript fully available?

Reviewer #1: Yes

Reviewer #2: No

4. Is the manuscript presented in an intelligible fashion and written in standard English?

Reviewer #1: Yes

Reviewer #2: Yes

5. Review Comments to the Author

Reviewer #1: I read carefully the manuscript entitle “Estimation of the fraction of COVID-19 infected people in U.S. states and countries worldwide”. In this work, the authors using a machine learning framework estimated the number of infected COVID-19 cases. This is a valuable work and could help policy makers for a better decision.

• Abstract: New York is 12% or 0.12%?

• Method: line 83: Infection-Fatality-Ratio is a better as this is not rate.

• For referencing some indicators like the time between onset of clinical sign to death or recovery and other indicators, I recommend to use systematic review and meta-analysis.

• The first paragraph of the results is better to move to the methods.

• The quality of the pictures and photos are very low and it is not clear the name of the countries. Please replace the photos with a photo with higher resolution.

• Please check all the text, you used the full format and abbreviation of some words in different parts of the text simultaneously. For example, you used Infection-Fatality-Rate (IFR).

Reviewer #2: This is an interesting study since this is a computational study and the first model estimating the actual fraction of currently infected people in each region. I would like to give some comments and questions related to the method and reporting of the article.

• The author did not write the paper systematically, such as paragraph from method is included in the introduction, and vice versa.

• We suggest the author to put sentences on page 4 line 67-68 into introduction section to emphasize the novelty of the study. The current issue and condition as the background of the study (including line 85-95 on page 5) should be included in introduction section, not in method section.

• On Page 3, line 49-51, the author stated “more importantly, we still do not know how many individuals are currently infected in many countries and regions”. Please add more references to strengthen the sentence.

• The Method section should be described in enough detail, so that someone else could follow the steps and replicate them if they wanted to do the same study.

• It would be better if the author put sentences on Page 3 line 54-65 in the method section regarding the source of data.

• The author should put the type or design or the study clearly in method section, can be written in the beginning of the first paragraph in the method section. It may also be briefly mentioned in the introduction section.

• It would be better if you mention and explain about spearman rank analysis, since variables were analyzed with spearman rank.

• The author mentioned 50 countries included in the development of the computation approach, however in the result and discussion section, the author talked mainly in the USA. Please comment further on other countries, too.

• Please add the reference for the sentence on page 8 line 177-179. The word “allegedly” may suggest assumption, not based on valid statistical data.

• The applicability of the study result needs to be stated in discussion section.

• In the discussion section, the author did not clearly state the strength and limitation of the study. Please describe efforts to overcome the limitation as well.

• The references were not based on Vancouver style, especially no 1 and 6. Please revise in regards to citation.

6. PLOS authors have the option to publish the peer review history of their article (what does this mean?). If published, this will include your full peer review and any attached files.

Reviewer #1: **Yes: **Hamid Sharifi

Reviewer #2: No

---

## [Author Response · Author response to Decision Letter 0]

18 Jan 2021

Response to reviewers was uploaded in a separate file.

---

## [Editor Report · Decision Letter 1]

26 Jan 2021

Estimation of the fraction of COVID-19 infected people in U.S. states and countries worldwide

PONE-D-20-32239R1

Dear Dr. Noh,

We’re pleased to inform you that your manuscript has been judged scientifically suitable for publication and will be formally accepted for publication once it meets all outstanding technical requirements.

Kind regards,

Yury E Khudyakov, PhD

Academic Editor

PLOS ONE
---

## [Editor Report · Acceptance letter]

29 Jan 2021

PONE-D-20-32239R1 

Estimation of the fraction of COVID-19 infected people in U.S. states and countries worldwide 

Dear Dr. Noh:

I'm pleased to inform you that your manuscript has been deemed suitable for publication in PLOS ONE. Congratulations! Your manuscript is now with our production department. 

Kind regards, 

on behalf of

Dr. Yury E Khudyakov 

Academic Editor

PLOS ONE